# Mechanical Chiseling and the Cover Crop Effect on the Common Bean Yield in the Brazilian Cerrado

Vagner do Nascimento [1], Orivaldo Arf [2], Marlene Cristina Alves [3], Epitácio José de Souza [4], Paulo Ricardo Teodoro da Silva [5], Flávio Hiroshi Kaneko [6], Arshad Jalal [3], Carlos Eduardo da Silva Oliveira [3], Michelle Traete Sabundjian [7], Samuel Ferrari [1], Fernando de Souza Buzo [2] and Marcelo Carvalho Minhoto Teixeira Filho [3,*]

[1] Department of Plant Production, Faculty of Agricultural and Technological Sciences, São Paulo State University (UNESP), Dracena 17900-000, SP, Brazil; vagner.nascimento@unesp.br (V.d.N.); samuel.ferrari@unesp.br (S.F.)

[2] Food Technology and Social Economy, Faculty of Engineering, Department of Crop Science, São Paulo State University (UNESP), Ilha Solteira 15385-000, SP, Brazil; o.arf@unesp.br (O.A.); fernando.buzo@unesp.br (F.d.S.B.)

[3] Rural Engineering and Soils (DEFERS), Department of Plant Protection, Faculty of Engineering, São Paulo State University (UNESP), Ilha Solteira 15385-000, SP, Brazil; marlene.alves@unesp.br (M.C.A.); arshad.jalal@unesp.br (A.J.); ces.oliveira@unesp.br (C.E.d.S.O.)

[4] Faculty Unibras of Goiás, Rio Verde 75909-310, GO, Brazil; epitaciosouza@fev.edu.br

[5] Precision Agriculture, Chapadão Foundation, Chapadão do Sul 79560-000, MS, Brazil; pauloteodoro@agronomo.eng.br

[6] Agronomy, Campus of Iturama, Federal University of Triangulo Mineiro (UFTM), Iturama 38280-000, MG, Brazil; flavio.kaneko@uftm.edu.br

[7] Department of Plant Science, Faculty of Social and Agrarian Sciences of Itapeva, Itapeva 18409-010, SP, Brazil; michelle.traete@professor.fait.edu.br

* Correspondence: mcm.teixeira-filho@unesp.br

**Abstract:** Core Ideas: (1) Superficial soil compaction in a no-tillage system. (2) Cultivation of cover crops in succession with annual crops. (3) Soil decompression with cultivation of the predecessor soil cover. (4) Unpacking soil with mechanical chiseling. (5) Biological chiseling with the cover crop effect on the common bean yield. Mechanical soil intervention with a chisel in cover crops (CC) is a promising alternative strategy to minimize superficial compaction of soil in a no-tillage system (NTS) of the Brazilian Cerrado. Thus, the objective of the current study was to evaluate the effects of mechanical chiseling associated with successor and predecessor cover crops on agronomic components and the grain yield of the common bean in NTS for two consecutive years. The experiment was designed in randomized blocks in a $5 \times 2$ factorial scheme with four replications. The treatments consisted of five cover crops (*Cajanus cajan*, *Crotalaria juncea*, *Urochloa ruziziensis* and *Pennisetum glaucum* and fallow), associated or not with soil mechanical chiseling. The results indicated that cultivation of *C. juncea* and *U. ruziziensis* as cover crops increased the initial and the final plant population and the number of pods plant$^{-1}$ of the common bean. The cultivation of *P. glaucum* as a predecessor crop with chiseling was observed with greater shoot dry matter and a greater number of grains pod$^{-1}$ and plant$^{-1}$ of the common bean while *C. cajan* and *C. juncea* have increased leaf N content in the common bean. The predecessor crops of *C. juncea* and *P. glaucum* with chiseling increased the grain yield of the "winter" common bean in succession. Therefore, cultivation of *C. juncea* and *P. glaucum* as predecessor crops along with chiseling are considered a sustainable strategy for improving the growth and the yield of successive crops in a no-tillage system of the Brazilian Cerrado.

**Keywords:** *Phaseolus vulgaris* L.; *Pennisetum glaucum*; *Crotalaria juncea*; soil compaction; *Urochloa ruziziensis*; green manure

## 1. Introduction

Common bean (*Phaseolus vulgaris* L.) is a grain crop of great economic and social relevance in the production systems of Brazil and the world, generating employment and income for those involved in its production chain. The common bean is a rich source of proteins, carbohydrates, vitamins, minerals, and fibers as well as being an important food for Brazilians [1,2]. The common bean was cultivated in an area of 2.93 million hectares with a production of 3.06 million tons and an average productivity of 1.033 kg ha$^{-1}$ in 2021–2022 in Brazil [3].

Soil compaction in a surface soil layer in a no-tillage system (NTS) is mainly due to the traffic of implements and machinery in soils having a high moisture content. The excess intervention of machinery in the absence of adequate agricultural planning for cultivation of cover or successive crops contributed to the intensification of compacted surface soil layers in NTS that could disrupt the soil structure, affecting soil fertility and crop productivity [4].

Mechanical intervention is carried out by means of scarifiers or subsoilers with cutting discs in front of stems that resist crop residues to incorporate into the soil. However, the long-term effects of mechanical soil scarification are different and changeable, ranging from a few months [5–7] to a few years [8,9], relying on the redisposition of soil particles as a result of soil type, weather conditions, machines and implement intervention, and predominant management practices, in particular a production system. Soil compaction has a direct effect on physical and mechanical characterizations of soil, which consequently impair plant growth and development. In general, soil compaction impairs water and nutrient uptake by limiting root length and penetration that all ultimately lead to poor plant growth and yield [10].

Cultivation of cover crops alone or in an intercropping system is a promising alternative to increase plant biomass productivity and nutrient accumulation in NTS [11–13]. The leftover straw of cover crops on the soil surface create a physical barrier between machinery tire and the soil surface to minimize surface compaction [14]. The plants of the Poaceae family, such as *Pennisetum glaucum, Urochloa ruziziensis* and *U. brizantha*, are fast-growing species that are capable of greater biomass production as well as promoting nutrient cycling [11,13]. The inclusion of Fabaceae species in production systems, either alone or in association with Poaceae, has increased the productivity of crops in succession by increasing nitrogen (N) availability [13] as results of biological nitrogen fixation (BNF) and a low Carbon/Nitrogen (C/N) ratio in straw [15,16].

Considering the research gap, the current study was hypothesized that mechanical soil scarification associated with predecessors of cover crops may improve the agronomic components and the grain yield of the common bean. In this context, the objective of the current study was to evaluate the effect of mechanical soil scarification associated with successive crops and predecessors of cover crops on agronomic and productive components and the grain yield of the common bean in the Cerrado region, being under NTS for twelve years in a low-altitude of Brazil.

## 2. Materials and Methods

### 2.1. Characterization and Experimental Conduct

The current research was developed in 2013 and 2014 in the Research and Extension Farm of the Sao Paulo State University, Selvíria, state of Mato Grosso do Sul, Brazil. The experimental site was located at 51°22′ West longitude and 20°22′ South latitude with an altitude of 335 m above sea level in a Rhodic Haplustox soil with a clayey texture [17].

The site received as annual average rainfall, temperature and relative humidity of 1370 mm, 23.5 °C, and 75%, respectively. The climate of the region was classified as Aw-type according to the Koppen climate classification, characterized as humid tropical with a rainy season in summer and a dry in winter. The climatic data recorded during experiments are shown in Figure 1. Irrigation was carried out by a central pivot sprinkler irrigation system at a water depth of 14 mm after every three days or according to crop requirements.

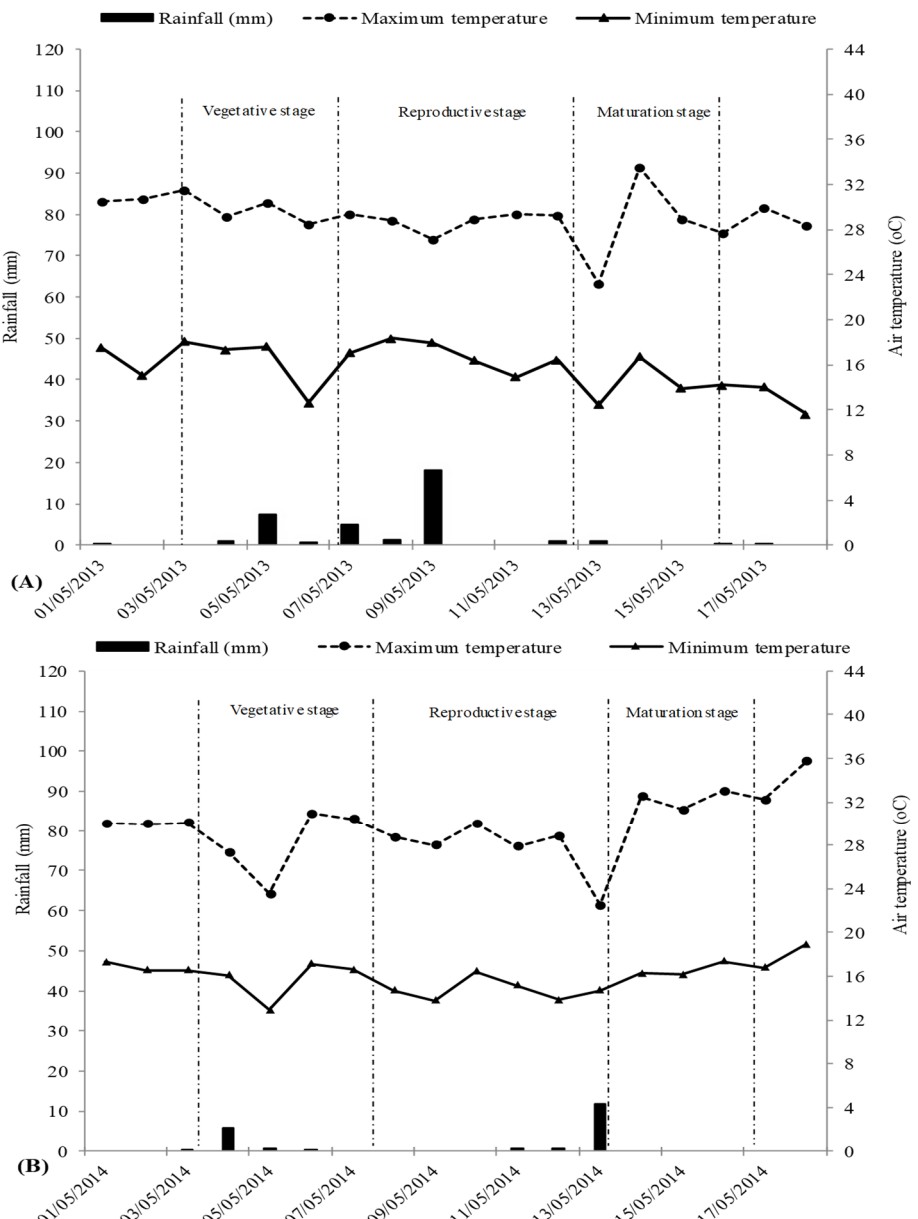

**Figure 1.** Rainfall (mm), maximum and minimum temperatures (°C) during winter bean cultivation, Selvíria, MS- Brazil in 2013 (**A**) and 2014 (**B**) harvests.

The area of the experiment has been cultivated in NTS for almost twelve years. The proposed cover crops and the upland rice were sown in succession in the 2012/13 and the 2013/14 cropping seasons.

### 2.2. Treatments and Experimental Design

The experiments were designed in randomized blocks in a 5 × 2 factorial scheme with four replications. The treatments consisted of five cover crops (*Cajanus cajan, Crotalaria juncea, Urochloa ruziziensis* and *Pennisetum glaucum* and fallow), with or without mechanical soil scarification. The fallow treatments with and without mechanical soil scarification were allowed to develop with spontaneous vegetation of predominant weeds species such as *Ipomoea acuminata, Bidens pilosa, Leonotis nepetaefolia, Conyza* spp., *Commelina benghalensis* and *Zea mays* (voluntary corn). Each experimental unit had a total area of 12.0 × 7.0 m and a useful area of 10.0 × 5.0 m.

The granulometry soil analysis of the research site were 385, 120, and 495 g kg$^{-1}$ of sand, silt, and clay, respectively, with a bulk density of 1.54 Mg m$^{-3}$ in 0.00–0.20 m layer (Table 1), according to Embrapa [18]. The chemical attributes of the soil were determined before installation of the experiment by collecting 20 random samples to form a composite sample in the entire experimental area in a 0.00–0.20 m layer presented the following results: 25 mg dm$^{-3}$ of P (resin); 16 g dm$^{-3}$ of OM; 4.7 pH (CaCl$_2$); K$^+$, Ca$^{2+}$, Mg$^{2+}$, H+Al, SB and CTC = 1.6, 13.5, 9.5, 35.5, 24.6, and 60.1 mmol$_c$ dm$^{-3}$, respectively, and 41% base saturation following the methodology of Raij et al. [19]. Based on the soil analysis and the recommendation of Raij et al. [19], the entire experimental area was applied 10 July 2012 with a dose of 1.6 kg ha$^{-1}$ dolomitic limestone (CaCO$_3$), having an effective neutralizing power of 85% through broadcast distribution to raise the base saturation to 70% for soil corrections.

**Table 1.** Physical attributes of the soil at different depths of the experimental area, before the installation of the experiment. Selvíria, Brazil, 2012.

| Depth (m) | Sandy | Silt | Clay | Macro | Micro | Total P. | SD |
|---|---|---|---|---|---|---|---|
| | g kg$^{-1}$ | | | m$^3$ m$^{-3}$ | | | Mg m$^{-3}$ |
| 0.00–0.05 | 403 | 157 | 440 | 0.08 | 0.36 | 0.44 | 1.49 |
| 0.05–0.10 | 389 | 127 | 484 | 0.06 | 0.35 | 0.41 | 1.56 |
| 0.10–0.20 | 385 | 120 | 495 | 0.07 | 0.35 | 0.42 | 1.54 |
| 0.20–0.40 | 352 | 121 | 527 | 0.10 | 0.36 | 0.46 | 1.42 |

Macro: Macroporosity; Micro: Microporosity; Total P.: total porosity; SD: Soil density, determined according to Embrapa's methodology (1997) [18].

The mechanical soil scarification was carried out on 9 August 2012. The slanted seven-shank scarifier (three on the front bar and four on rear) and a chisel tip with a spacing between 300 mm, an attack angle of 22°, and a crushing roller coupled to the tractor traction bar were used before cover crop cultivation. The average working depth was set to 0.25 m and the cutting swath width was 2.10 m. The operations were performed at a soil moisture content close to the friability point as well as a disk harrow was carried out in chiseled plots.

All cover crops were sown on 14 August 2012 and 5 September 2013 by a manual seeding machine with 0.45 m row spacing and without mineral fertilizer application. The sowing density of 60 kg ha$^{-1}$ for *C. cajan*, 30 kg ha$^{-1}$ for *C. juncea*, 12 kg ha$^{-1}$ for *U. ruziziensis*, and 12 kg ha$^{-1}$ for *P. glaucum* was maintained during cultivation.

All cover crops and fallows were desiccated at 68 days after sowing (DAS) in 2012 and at 63 DAS in 2013 with the application of herbicide glyphosate (1.440 g ha$^{-1}$ of a.i.) + 2,4-D (670 g ha$^{-1}$ of a.i.). The herbicides were applied with a tractor-trailer sprayer at a 200 L ha$^{-1}$ application rate. After 10 days of desiccation, crop residue management was carried out in all treatments with the help of a horizontal mechanical straw crusher at a cutting height of 0.10 m above ground level.

Common beans as a test crop were sown on 3 May 2013 and 13 May 2014 by using a drill sowing method, following the same experimental design and treatment combination. Seeds were treated with carboxin + tiram (50 + 50 g a.i. per 100 kg of seeds) + 500 g of peaty inoculant with *Rhizobium tropici* per 50 kg before installation of the experiment into the field. The common bean cultivar "Pérola" was used in both years' cultivation, which had type III plants (indeterminate growth habit) and carioca type grains (seed coat with brown stripes and represent almost 75% share in market). This cultivar is mostly used from 1995 to 2018 in Brazil. It has semi-erect plants with high productivity, excellent grain formation, and a short cycle, which is ideal for winter cultivation. The seed planter was adjusted to a distribution of 12 plants per meter after emergence. Each common bean plot consisted of 14 rows 0.45 m apart and 12 m long. The useful area in each plot was consisted of 12 central lines with 0.5 m neglected at both ends of each line. The plants were uniformly emerged at 6 DAS in both growing years.

The fertilization was performed in sowing furrows according to the chemical characteristics of the soil and the recommendations of Ambrosano et al. [20]. The dose of

250 kg ha$^{-1}$ of NPK (04-30-10) + 0.3% Zn in 2013 and 220 kg ha$^{-1}$ of NPK (08-28-16) in 2014 were applied in the sowing of the common bean. Nitrogen fertilization (60 kg ha$^{-1}$) in topdressing was carried out at 19 and 21 days after emergence (DAE) in 2013 and 2014, respectively, from the source of ammonium sulfate (20% N and 22% S) in 2013 and urea (45% N) in 2014.

Weeds were controlled with herbicides applied by tractor sprayer at a flow rate of 200 L ha$^{-1}$. The entire experimental area was desiccated with the application of glyphosate (1.440 g ha$^{-1}$ of a.i.) and bentazon (720 g ha$^{-1}$ of a.i.) 12 days before plantation on 26 April 2013. In addition, desiccation of the experimental area was performed on 29 April 2014 with the application of glyphosate (1.440 g ha$^{-1}$ of a.i.) + carfentrazone-ethyl (20 g ha$^{-1}$ of a.i.), 13 days before installation of the common bean experiment. The post-emergence weeds were controlled with the herbicides fenoxaprope-P-ethyl (83 g ha$^{-1}$ of a.i.) and bentazon (720 g of ha$^{-1}$ of a.i.) applied at 15 and 23 DAE, respectively. The remaining weeds were eliminated manually with the aid of a hoe. The other crop management and agronomic practices were uniformly performed in all treatments as recommended for common bean crop in the region. The full flowering was initiated at 40 and 42 DAE, while the crop was manually harvested at 96 and 94 DAE in 2013 and 2014, respectively.

### 2.3. Assessments

The evaluations recorded were: (a) initial and final plant population at 8 DAE and at the time of harvest, respectively. Number of plants were counted in two central lines of each plot in order to calculate initial population and final population of plants ha$^{-1}$; (b) shoot dry matter was determined by collecting 10 plants from central lines at the full flowering (R6) stage, placed in identified paper bags and subjected to drying in a forced airtight oven at a temperature of 65 °C until reaching a uniform weight. Then, the samples were weighed and converted into g plant$^{-1}$; (c) 10 random trifoliate leaves were collected during the R6 stage, dried in an airtight oven at a temperature of 65 °C, ground in a Wiley mill, and subjected to sulfuric digestion for leaf nitrogen content according to methodology of Malavolta et al. [21]; (d) production components: 10 plants were collected from the useful area of each plot at the time of harvest to evaluate the number of pods plant$^{-1}$, number of grains plant$^{-1}$, number of grains pod$^{-1}$; (e) a mass of 100 grains was calculated by counting two random samples of 100 grains in each plot and weighed for hundred grains plot$^{-1}$; (f) the grain yield was determined by harvesting three central rows in each plot and then left in the full sun to dry. Each sample was separately submitted to mechanical threshing, weighed, and transformed into kg ha$^{-1}$ (13% wet basis).

### 2.4. Statistical Analysis

After verifying normality, the data were submitted to analysis of variance (F test) and means were compared by Tukey test at 5% of significance for cover crops (CC) and use or not of chiseling (MSC). Means of significant interaction between sources of variations (CC and MSC) were also compared by Tukey test 5% probability of significance according to Pimentel et al. [22]. The statistics were performed by using Sisvar 5.6 [23].

## 3. Results

The interactions of mechanical scarification and cover crop cultivation were significant for shoot dry matter and leaf nitrogen content (LNC) in 2013 and 2014 as well as for the initial plant population in 2013 (Table 2). The isolated cultivation of *C. juncea* as a cover crop increased the initial plant population in relation to *P. glaucum*. The treatments effects for the initial plant population in 2014 and the final plant population in 2013 and 2014 were not significant (Table 2). The initial plant population in 2014 was increased within treatments of chiseling in cultivation of *P. glaucum.* The final plant populations in 2013 and 2014 were increased with chiseling and fallow treatments. In general, the initial plant population of the present study was an adequate population for the common bean cultivar Pérola.

**Table 2.** Mean values of initial and final plant population, shoot dry matter, and leaf nitrogen (N) content of "winter" common bean after mechanical decompaction and predecessor cover crops, MS-Brazil, 2013 and 2014.

| Treatments | Plant Population at $V_2$ Stage (Initial) | | Plant Population at Harvest (Final) | | Shoot Dry Matter | | Leaf N Content | |
|---|---|---|---|---|---|---|---|---|
| | Plant $ha^{-1} \times 1000$ | | | | $g\ plant^{-1}$ | | $g\ kg^{-1}$ | |
| | 2013 | 2014 | 2013 | 2014 | 2013 | 2014 | 2013 | 2014 |
| **Mechanical Soil Chiseling (MSC)** | | | | | | | | |
| Without | 219 | 238 | 127 | 161 | 9.9 | 11.10 | 37.27 | 44.25 |
| With | 219 | 245 | 131 | 155 | 10.5 | 12.90 | 39.69 | 42.83 |
| **Cover crops (CC)** | | | | | | | | |
| Fallow | 221 b | 238 | 136 | 164 | 9.5 | 11.71 | 38.88 | 41.72 |
| *U. ruziziensis* | 217 b | 244 | 131 | 155 | 10.8 | 13.74 | 37.40 | 44.18 |
| *C. juncea* | 226 a | 231 | 130 | 164 | 10.9 | 12.19 | 38.11 | 44.18 |
| *C. cajan* | 221 b | 242 | 121 | 145 | 9.2 | 12.19 | 40.16 | 44.80 |
| *P. glaucum* | 210 c | 253 | 128 | 163 | 10.8 | 10.18 | 37.84 | 41.28 |
| **F values** | | | | | | | | |
| MSC | 0.02 ns | 1.29 ns | 1.64 ns | 1.62 ns | 5.85 * | 24.99 * | 18.49 * | 4.06 ns |
| CC | 22.33 * | 1.22 ns | 2.15 ns | 2.20 ns | 7.86 * | 9.97 * | 2.98 * | 3.38 ns |
| MSC $\times$ CC | 10.53 * | 0.35 ns | 0.19 ns | 1.22 ns | 7.98 * | 6.40 * | 2.52 * | 3.65 * |
| CV(%) | 1.60 | 8.72 | 8.50 | 10.17 | 8.02 | 9.50 | 4.62 | 5.10 |

ns non-significant and * significant at 5% probability by F test. Means followed by same letter for scarification and cover crops did not differ statistically from each other by Tukey test at 5% significance.

The interactions of chiseling and the cover crop in 2013 and 2014 were significant (Figure 2A,B). A greater shoot dry matter of the common bean was observed with mechanical scarification in *P. glaucum*, which was statistically similar with fallow, *C. juncea* and *U. ruziziensis* cultivation in the 2013 cropping season. A lower shoot dry matter was noted at scarification within *C. cajan*, which was statistically similar with fallow without scarification (Figure 3A). The cultivation of *C. juncea* in non-chiseled treatments corresponded to greater shoot dry matter, which was statistically at per with *U. ruziziensis* and *C. cajan* in 2013.

In 2014, a greater shoot dry matter of the common bean was recorded with chiseling in cultivation of *U. ruziziensis* as cover crop in NTS. The cultivation of *C. juncea* in non-chiseled treatments were observed with greater shoot dry matter production which was statistically not different from all other cover crop cultivation and fallow treatments except *P. glaucum* in non-scarified areas, which produced a lower shoot dry matter of the common bean in 2014 (Figure 3B).

The interaction of soil chiseling within each vegetation of cover crop for leaf N content (LNC) was significant in 2013 and 2014 (Figure 2C,D). Soil chiseling within cultivation of *C. cajan* increased leaf N content, which was statistically similar with *C. juncea* and fallow in 2013. Higher leaf N content in non-chiseled treatments was observed within fallow treatments. The lowest leaf N content was noted with cultivation of *C. juncea* in non-chiseled treatments in 2013 as compared with all other treatments within chiseling and without chiseling (Figure 2C).

The cultivation of cover crops within mechanical chiseling in 2014 also showed differences for leaf N content (Figure 2D). A higher leaf N content was noted with *C. juncea* in scarified soil which was statistically at per the treatments previously cultivated with *C. cajan* and *P. glaucum*. The lowest leaf N content in 2014 was noted within fallow in chiseling treatments as compared to other treatments. The previously cultivated *U. ruziziensis* within non-chiseled plots were observed with a higher leaf N content, which was statistically not different from the cultivation of *C. cajan* within non-chiseled treatments in 2014 as compared to other treatments. The lowest leaf N content was observed within cultivation of *P. glaucum* at without soil chiseling treatments (Figure 2D).

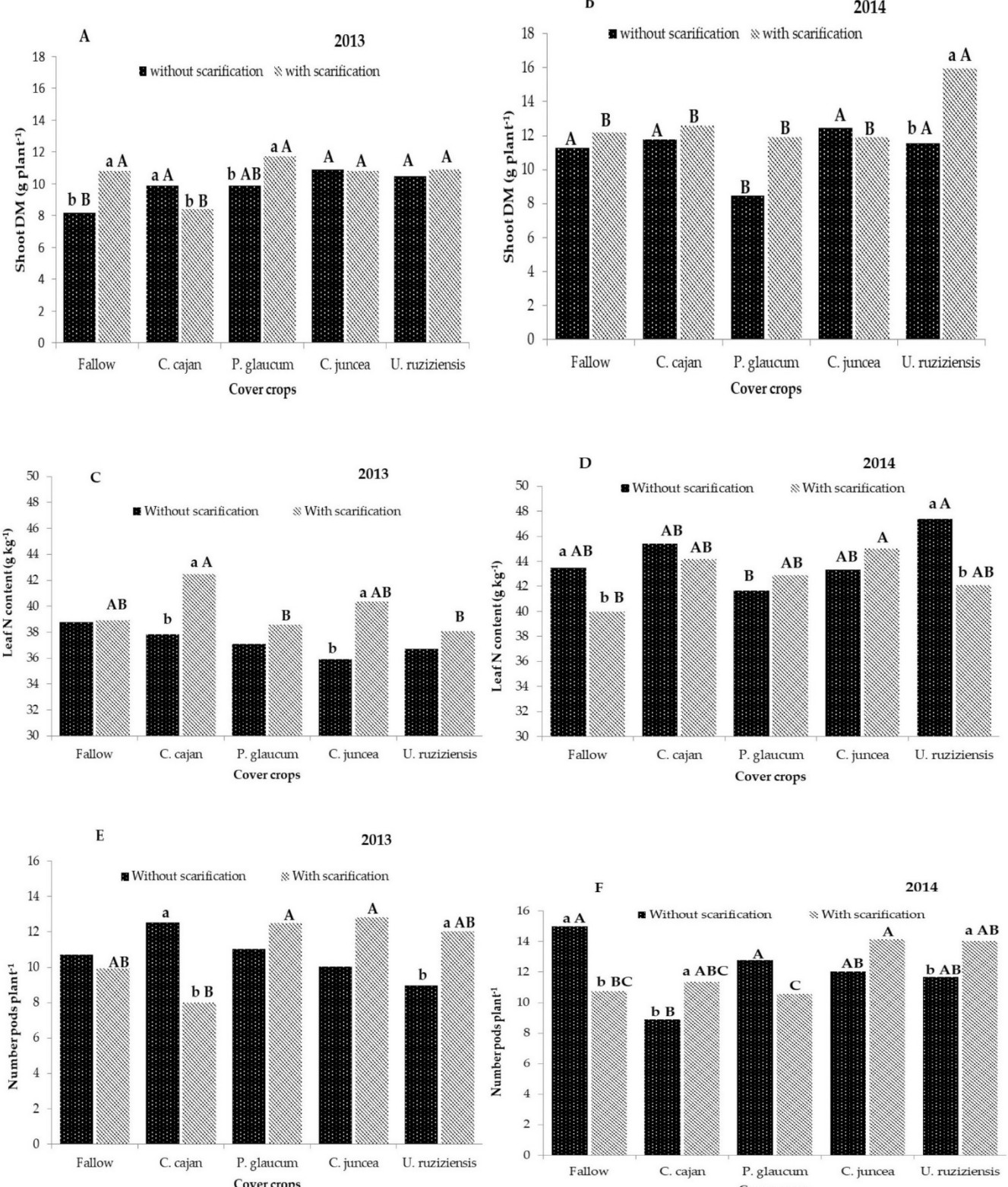

**Figure 2.** Interactions for shoot dry matter (DM- **A**,**B**), leaf nitrogen (N) content (**C**,**D**) and number of pods per plant of "winter" common bean (**E**,**F**)after mechanical soil chiseling (MSC) and predecessor cover crops (CC) in a no-tillage system. Averages followed by same lowercase letter for CC within MSC and capital for MSC within CC did not statistically differ by Tukey test at 5% significance in 2013 and 2014, respectively.

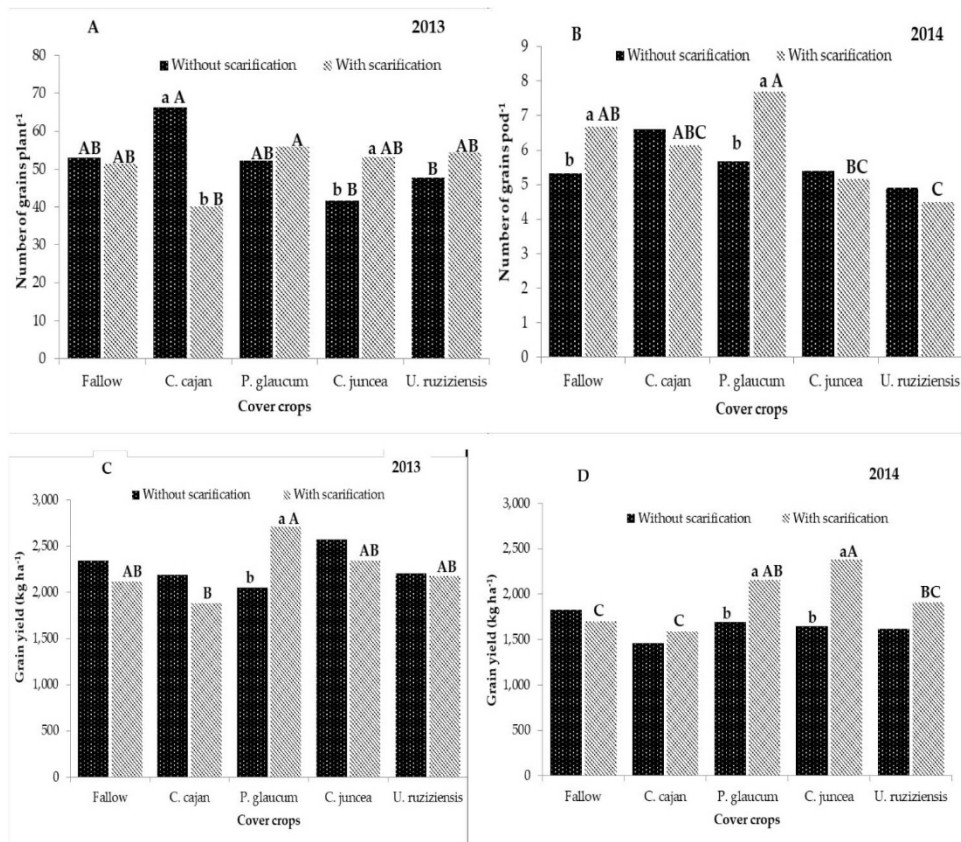

**Figure 3.** Significant interactions for number of grains per plant in 2013 (**A**), number of grains per pod in 2014 (**B**) and grain yield in 2013 and 2014 (**C,D**) of "winter" common bean after mechanical soil chiseling (MSC) and predecessor cover crops (CC) implanted in NTS. Averages followed by the same lowercase letter for CC within MSC and capital letters for MSC within CC did not statistically differ by Tukey test at 5% significance, respectively.

There were significant interactions of mechanical soil chiseling and cover crop cultivation for a number of pods plant$^{-1}$ (NPP) in 2013 and 2014, number of grains plant$^{-1}$ (NGP) in 2013, and number of grains pod$^{-1}$ in 2014 (Table 3).

The number of pods plant$^{-1}$ (NPP) in 2013 increased with the incorporation of *C. juncea* as a cover crop in the area with chiseling, which was statistically not different from the treatments with predecessor cultivation of *P. glaucum* and *U. ruziziensis*. However, a reduced NPP was noted in soil chiseling within previously cultivated *C. canjan* treatments (Figure 2E). In addition, a greater NPP without soil chiseling were observed with incorporation of *C. canjan* as compared to other treatments. The lowest NPP in 2013 were noted with predecessor cultivation of *U. ruziziensis* in non-chiseled treatments (Figure 2E).

The interactions of chiseling within each cover crop in 2014 verified that the number of pods plant$^{-1}$ were higher in chiseling with *C. juncea* and *U. ruziziensis*. The lowest number of pods plant$^{-1}$ were noted with predecessor cultivation of *P. glaucum* in soil chiseling (Figure 2F). The treatments without chiseling in fallow produced a higher number of pods plant$^{-1}$, which was statistically not different from all other cover crop cultivation in non-chiseled plots except treatments with *C. cajan* in 2014 (Figure 2F).

The interaction of cover crop in predecessor and chiseling for number of grains plant$^{-1}$ (NGP) in 2013 was significant (Figure 3A). The interaction exhibited that NGP increased with the incorporation of *P. glaucum* residues, which was statistically similar with *C. juncea* and *U. ruziziensis* with soil chiseling. The lowest NGP was observed with *C. cajan* residue incorporation in soil chiseling treatments (Figure 3). The cultivation and the incorporation of *C. cajan* without scarification produced a higher NGP, while *C. juncea* and *U. ruziziensis* corresponded to a lower NGP in the common bean in 2013 (Figure 3A).

**Table 3.** Mean values of number of pods and grains plant$^{-1}$ and number of grains pod$^{-1}$ of "winter" common beans after mechanical chiseling and predecessor cover crops, Selvíria, MS, Brazil, 2013 and 2014.

| Treatments | Number of Pods Plant$^{-1}$ | | Number of Grains Plant$^{-1}$ | | Number of Grain Pod$^{-1}$ | |
|---|---|---|---|---|---|---|
| | 2013 | 2014 | 2013 | 2014 | 2013 | 2014 |
| **Mechanical Soil Chiseling (MSC)** | | | | | | |
| Without | 10.66 | 12.07 | 52.19 | 66.56 | 4.67 | 5.58 |
| With | 11.06 | 12.18 | 51.02 | 71.45 | 4.75 | 6.04 |
| **Cover crops (CC)** | | | | | | |
| Fallow | 10.33 | 12.87 | 52.17 | 75.40 | 4.86 | 6.00 |
| *U. ruziziensis* | 10.50 | 12.86 | 51.01 | 59.53 | 4.55 | 4.70 |
| *C. juncea* | 11.43 | 13.08 | 47.53 | 69.10 | 4.71 | 5.28 |
| *C. cajan* | 10.28 | 10.13 | 53.21 | 64.24 | 4.83 | 6.38 |
| *P. glaucum* | 11.77 | 11.68 | 54.08 | 76.76 | 4.59 | 6.69 |
| **F values** | | | | | | |
| MSC | 0.39 ns | 0.04 ns | 0.25 ns | 1.66 ns | 0.33 ns | 2.55 ns |
| CC | 0.93 ns | 4.76 * | 0.96 ns | 2.96 ns | 0.77 ns | 6.58 * |
| MSC × CC | 4.83 * | 7.47 * | 7.99 * | 0.97 ns | 0.22 ns | 3.30 * |
| CV(%) | 18.66 | 13.30 | 14.28 | 17.42 | 9.14 | 15.45 |

ns not significant and * significant at 5% significance level by F test. Means followed by same letter for scarification and CC did not statistically differ from each other by Tukey test at 5% significance.

The number of grains pod$^{-1}$ in 2013 were not significantly influenced by soil chiseling and the cover crop cultivation, nor by their interaction (Table 3). In addition, a number of grains pod$^{-1}$ in 2014 were positively influenced by soil management and cover crops and their interaction (Figure 3B). The number of grains pod$^{-1}$ increased within treatments of soil chiseling, which was statistically similar to fallow and the incorporation of *C. cajan* treatments. However, a lower number of grains pod$^{-1}$ were noted within *U. ruziziensis* incorporation and soil chiseling in 2013 as compared to other cover crops (Figure 3B). In addition, the cultivation of *C. canjan* without soil chiseling produced a higher number of grains pod$^{-1}$, while cultivation of *U. ruziziensis* without soil chiseling produced a lower number of grains pod$^{-1}$ in 2013 (Figure 3B).

The 100-grains mass (HGM) of the common bean was positively influenced by mechanical soil chiseling in 2014 with an increase of 4.55% in relation to without chiseling (Table 4).

Grain yield (GY) of common bean in 2013 and 2014 were significantly influenced by cover crops and mechanical soil chiseling (Table 4). The interactions for grain yield were also significant (Figure 3C,D). Grain yield of the common bean in 2013 increased with cover crops *P. glaucum*, which was statistically similar to treatments incorporated with *C. juncea* and *U. ruziziensis* under soil mechanical chiseling (Figure 3C). The lowest GY in 2013 under soil chiseling was noted within residues incorporation of *C. cajan* in relation to other treatments (Figure 3C). In 2014, grain yield increased with *C. juncea* incorporation and soil mechanical chiseling, which was statistically at per with incorporation of *P. glaucum*. In addition, treatments that incorporated *P. glaucum* and *C. juncea* were observed with greater GY under soil without mechanical chiseling in relation to other cover crops while *C. cajan* was noted with lower GY without mechanical chiseling treatments (Figure 3D).

**Table 4.** Mean values of 100 grains mass and grain yield of "winter" common beans after mechanical decompaction and predecessor cover crops in 2013 and 2014.

| | 100 Grains Mass | | Grain Yield | |
|---|---|---|---|---|
| | 2013 | 2014 | 2013 | 2014 |
| | g | | kg ha$^{-1}$ | |
| **Mechanical Soil Chiseling (MSC)** | | | | |
| Without | 28.69 | 25.27 b | 1.649 | 2.273 |
| With | 29.28 | 26.42 a | 1.947 | 2.247 |
| **Cover crops (CC)** | | | | |
| Fallow | 28.85 | 24.99 | 1.761 | 2.230 |
| *U. ruziziensis* | 29.53 | 25.99 | 1.763 | 2.192 |
| *C. juncea* | 29.17 | 26.76 | 2.015 | 2.456 |
| *C. cajan* | 28.20 | 26.23 | 1.525 | 2.037 |
| *P. glaucum* | 29.19 | 25.26 | 1.925 | 2.383 |
| **F values** | | | | |
| MSC | 1.78 [ns] | 7.90 * | 18.97 * | 0.08 [ns] |
| CC | 1.08 [ns] | 2.47 [ns] | 6.02 * | 2.58 [ns] |
| MSC × CC | 2.59 [ns] | 1.78 [ns] | 4.56 * | 3.74 * |
| CV(%) | 4.43 | 7.10 | 12.01 | 12.86 |

[ns] not significant and * significant at 5% significance level by F test. Means followed by same letter for scarification and CC did not statistically differ from each other by Tukey test at 5% significance.

## 4. Discussion

The soil with superficial mechanical chiseling provided a greater development of the common bean in all predecessor cover crop cultivation. This is due to soil chiseling as a result of mechanical operation that allowed greater root development of crops in sequence [12]. Since, the current experimental site was under a no-tillage system for 12 years, accumulating successive soil particles with low porosity in the long term due to transit of machines in sowing, crop treatments, and harvesting. This high degree compaction of the soil surface layer may harm the soil profile, development of tap and secondary roots in depth, and also reduce water and nutrient absorption, which all lead to the limited development and productivity of crops of economic interest [13,24–26].

The positive results in the area cultivated with *P. glaucum* and *U. ruziziensis* may be related to the fasciculate and the fine root systems of these species. Galdos et al. [27] demonstrated that biopores of the Poaceae root system are not only improving soil physical structure but also reducing nitrate losses and increasing the stability of soil and plants in succession. These authors also indicated that finer roots have increased macro-porosity and connectivity between pores, while reducing nitrate leaching. The inclusion of cover crops in an agricultural system increases nutrient mineralization—especially N—as their residues increase absorption of this nutrient by the roots of the plants and transported it to leaf tissues [28]—especially when they have a low C/N ratio.

The higher leaf N content in the common bean under soil chiseled areas and *C. cajan* cultivation may be explained by the amount of dry matter and the low C/N ratio of fabacean crop residues that had rapidly decomposed and released to the soil surface but still lower as compared to other treatments in 2013 (Figure 2C). The plants absorbed the same amount of N, but distributed it in a smaller amount into plant tissues, which led to lower growth [29]. There were differences in both of the studied years but still all observed leaf N contents were within the appropriate range (24 to 52 g kg$^{-1}$ of N) for better development of common bean plants according to Malavolta et al. [30]. Some other studies using legumes as cover crops provided greater N accumulation to successive cultivated crops due to their ability of BNF that can increase soil availability of N for uptake by plant roots [31–33].

The leaf N content increased in *U. ruziziensis* and *C. canjan* species of Fabaceae in 2014 (Figure 2D). This may be due to their capability to perform BNF, which is the best alternative to providing more N to an agricultural system. In addition, a lower C/N ratio of fabaceous straw residues provided a rapid decomposition and release of nutrients to the plants grown in sequence without immobilization of N in a decomposition process [11,15,34,35]. The rapid decomposition and mineralization of legume cover crop residues increases the availability of N in the soil, ranging from 800 to 1200 mg of N $kg^{-1}$ dry mass of residues in legumes, while in grasses it is between 200 to 400 mg of N $kg^{-1}$ of dry mass of residue [33,36,37].

Cover crops such as *C. juncea* have the ability to fix N biologically with a high production of dry matter that persists in soil with a low C/N ratio [12]. All of these factors contributed to a greater growth of the common bean, setting off flowering and pods through incorporation of straw residues that reduces soil temperature variation, increases water retention, and maintains soil moisture long term, in addition to the cycling and the availability of N by biological fixation [34,38]. The results of *P. glaucum* in the present study were justified by the same factors as *C. juncea* except BNF. The straw production of *P. glaucum* has a slightly higher C/N ratio than *C. juncea*. In addition, this species has shown a greater potential for nutrients (P, K, Ca, Mg and S) cycling in soil [13,15,39].

The 100 grain mass was positively influenced by soil mechanical chiseling in 2014 (Table 4) with an increase of 4.55%. This may be due to the resistance of root penetration to the mechanical chiseling of soil, which improved the root system [40,41] with a greater absorption of water and nutrients [42,43]. The greater efficiency of soil exploitation by roots has a direct effect on the transport of nutrients and water from the roots to the aerial part of the plants, increasing photosynthesis and the transport of photo-assimilates to grains [44], leading to better grain filling.

The grain yield of the common bean was increased in the treatments with *C. juncea* and it was lowered with *C. cajan* and fallow under soil chiseling (Table 4; Figure 2C,D). In this case, the results may be the consequence of *C. juncea* incorporation which increased shoot DM, NGP, and pods plant$^{-1}$ of the common bean in this treatment. This is more beneficial in combination with mechanical chiseling, breaking the compacted soil superficial layers and improving the physical conditions of soil for the better development of the root system [5,26]. The integrated use of grasses as cover crops and chiseling provided greater soil exploration by soybean roots which has consequently increased grain yield [44]. The crop residues of *C. juncea* are very similar to Poaceae, with a relatively high persistence due to high lignin content in stems, a reduced soil temperature, and being able to minimize water loss, thus benefiting the root-shoot growth of crops in succession [13,40,41]. In addition, treatments with *C. cajan* and fallow were observed with a lower GY of the common bean. It may be due to lower straw production and persistence on the soil surface [13], which didn't support the development of plants as other cover crop species.

The greater GY with *P. glaucum* straw residues (Figure 3C,D) is due to the effect of these residues on the development of the common bean, increasing NGP, leaf N content, and shoot DM (Figure 2). As already demonstrated, these effects are due to a high production of DM, greater persistence of straw on soil, and nutrient cycling of this specific cover crop species [11,13,36]. In addition, *C. cajam* proved to be a cover crop with the lowest potential to improve the development of the common bean and productivity in mechanical soil chiseling. This effect can be explained by the fact that this species has a lower straw production and less persistence in soil due to its low C/N ratio and its small contribution to nutrient cycling [13,31]. The use of legumes as cover crops in successive years can reduce N fertilization by 23% [45] with uptake and transportation to shoots and grains of sorghum [46]. Pedrinho et al. [34] indicated that using grasses as cover crops has increased the grain yield of the common bean at doses of 100 and 150 kg $ha^{-1}$ of N fertilization, while using legumes as a predecessor cover plant has increased the grain yield of the common bean at a dose of 50 kg $ha^{-1}$ of N, with a reduction of 100 kg $ha^{-1}$ of N fertilization. Thus,

use of cover crops in agricultural systems are able to reduce nitrogen fertilizers and to increase profitability and economy.

## 5. Conclusions

The use of cover crops and chiseling in a no-tillage system has a positive impact on the agronomic and the productive characteristics of the winter common bean in succession. The cultivation of *C. juncea* and *U. ruziziensis* as cover crops increased the initial and the final plant population, and a number of pods plant$^{-1}$ of the common bean. The cultivation of *P. glaucum* as a predecessor crop with chiseling has improved shoot dry matter, the number of grain pod$^{-1}$ and plant$^{-1}$, and the grain yield of the common bean, while *C. cajan* and *C. juncea* has increased the leaf N content in this legume. The predecessor crops of *C. juncea* and *P. glaucum* with chiseling increased the yield of the "winter" common bean in succession.

Scarification increases soil exploitation by common bean roots and, with the use of *C. juncea* and *P. glaucum*, it improves the absorption of available nutrients in the cover crops residues for successive crops.

**Author Contributions:** Conceptualization, V.d.N. and O.A.; methodology, M.C.A.; software, A.J.; validation, P.R.T.d.S., F.H.K. and M.T.S.; formal analysis, V.d.N.; F.H.K. and C.E.d.S.O.; investigation, F.d.S.B. and M.C.M.T.F.; resources, V.d.N., M.C.A., S.F. and O.A.; data curation, M.C.M.T.F.; writing—original draft preparation, V.d.N. and A.J.; writing—review and editing, M.C.A., E.J.d.S., C.E.d.S.O. and M.C.M.T.F.; visualization, E.J.d.S. and M.T.S.; supervision, O.A.; project administration, V.d.N. and O.A.; funding acquisition, V.d.N. All authors have read and agreed to the published version of the manuscript.

**Funding:** From FAPESP and CNPq for financial support and for granting a doctoral scholarship to the first author by FAPESP, Process: 2012/05945-0. This research was financed in part by the Coordenação de Aperfeiçoamento de Pessoal de Nível Superior—Brasil (CAPES/AUXPE Award Number 88887.592666/2020-00 | 0242/2021).

**Institutional Review Board Statement:** Not applicable for studies not involving humans or animals.

**Informed Consent Statement:** Not applicable.

**Data Availability Statement:** Not applicable.

**Acknowledgments:** To FAPESP and CNPq for financial support and for granting a doctoral scholarship to the first author by FAPESP, Process: 2012/05945-0. To the UNESP, Faculty of Engineering, Ilha Solteira Campus for the infrastructure and human resources made available.

**Conflicts of Interest:** The authors declare no conflict of interest.

## Abbreviations

CC: cover crops, CV: coefficient of variation, DAE: days after emergence, DAS: days after sowing, BNF: biological nitrogen fixation, GM: hundred grains mass, DM: Dry matter, MSC: mechanical soil chiseling, NGP: number of grains per plant, GP: grains per pod, NPP: number of pods per plant, GP: grains productivity, NTS: no-tillage system, LNC: leaf nitrogen content.

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
