# Peer review of "Mechanical Chiseling and the Cover Crop Effect on the Common Bean Yield in the Brazilian Cerrado"

_agriculture, doi:10.3390/agriculture12050616_

Round 1

Reviewer 1 Report

The manuscript is well structured and the writing is very clear. My comments are the following. - I think that climatological graphs are not necessary in the text of the manuscript. They could go as annexes but I leave it to the choice of the authors. - Why were different fertilizers used the two years? I think it would be very interesting to do tests inoculating with rhizobia - Why do you use the Perola variety? Is it very used? What is it? Commercial variety, improved variety... The authors should put some more reference to the years 2020, 2021 and 2022 

Author Response

Response to Reviewer 1 Comments

Dear Reviewer,

We would like to express our gratitude for the reviewer who took the time to provide such a thorough review of our manuscript. We believe that the changes suggested have made our manuscript much more direct and much easier to follow.

Thanks!

Point 1: The manuscript is well structured and the writing is very clear. My comments are the following. - I think that climatological graphs are not necessary in the text of the manuscript. They could go as annexes but I leave it to the choice of the authors.

Response 1: The authors thanks to reviewer for appreciation of our manuscript. We believe that climate data is an important part of the study, to know the enivronmental conditions of growing crops. So we chose to keep it in the material and methods, as it is important information for local characterization.

Point 2: - Why were different fertilizers used the two years? I think it would be very interesting to do tests inoculating with rhizobia.

Response 2: The fertilization was consisted of 250 kg ha-1 of NPK (04-30-10) + 0.3% Zn in 2013 and 220 kg ha-1 of NPK (08-28-16) in 2014. The change of NPK fertilizer occurred because the P content of the soil was high and it was not necessary to have a high supply of P in the second year of cultivation, that is, the supply of P was reduced. In addition, the K content in soil in the 2013 crop was medium, thus fertilization was carried out with 25 kg of K ha-1 at sowing, due to the high removal of K by the plants, and in order to increase the K levels in the soil for the plants in the second year, a slightly higher potassium fertilization was carried out (with 35 kg of K ha-1 at sowing).

Nitrogen fertilization (60 kg ha-1) in topdressing was carried out at 19 and 21 days after emergence (DAE) in 2013 and 2014 respectively from the source of ammonium sulfate (20% N and 22% S) in 2013 and urea (45 % N) in 2014. In the first year of cultivation, Ammonium sulfate was used as a source of N in coverage to meet the needs of common bean plants in S, this supply was sufficient for the cultivation of the following year in a clayey soil.

The inoculation of beans with Rhizobia was not carried out, as this is not something necessary or mandatory as it is for soybean cultivation. Additionally, this inoculation with Rhizobium tropici was not performed to reduce the chances of obtaining better common bean production responses from the mineralization of cover crops (treatments).

Point 3: Why do you use the Perola variety? Is it very used? What is it? Commercial variety, improved variety...

Response 3: The common bean cultivar BRS Perola is the most used in Brazil from 1995 to 2018. This cultivar has semi-erect plants with high productivity, excellent grain formation and short cycle and also ideal for winter cultivation. For the information it is added in the Material and Methods section.

Point 4: The authors should put some more reference to the years 2020, 2021 and 2022

Response 4: The authors thanks to reviewer. The references are updated as suggestted and highlighted in red color in list of references.

Reviewer 2 Report

Dear Authors,
Extensive and serious effort is required to improve the language of this manuscript. Specific suggestions have been mentioned in the attached manuscript.
However, in general point-wise suggestions is given here:
1. The data used in this manuscript is very old. Incorporate some recent data information
2. Abstract is not well written. Add results with some data value and than overall conclusion.
3. Introduction must be improved by adding more justification of your study, hypothesis.
4. Materials and method- Improve figure and write your methodology of study properly. Add recent information/data and refurnish it again.
5. Result: Represent data through pooled analysis otherwise use years for statistical analysis and add recent data.
6. Rewrite conclusion in one paragraph in detail.

Overall, Refurnish your manuscript for good publication.

Author Response

Response to Reviewer 2 Comments

Dear Reviewer,

We would like to express our gratitude for the reviewer who took the time to provide such a thorough review of our manuscript. We believe that the changes suggested have made our manuscript much more direct and much easier to follow.

Thanks!

Point 1: The data used in this manuscript is very old. Incorporate some recent data information.

Response 1: The authors are thankful to reviewer. The manuscript is updated with new information as suggested.

Point 2: - Abstract is not well written. Add results with some data value and than overall conclusion.

Response 2: The abstarct is revised with most important results data and conclusions. Hope this version has met the expectatiosn of the reviewer.

Point 3: Introduction must be improved by adding more justification of your study, hypothesis.

Response 3: The introduction was improved with recent references, information and hypothesis and highlighted in red color in main text.

Point 4: Materials and method- Improve figure and write your methodology of study properly. Add recent information/data and refurnish it again.

Response 4: The information are updated as suggested and hightlighted in yellow color. 

Point 5: Result: Represent data through pooled analysis otherwise use years for statistical analysis and add recent data.

Response 5: Thanks. We appreciate your suggestion but at this point we are not agree with reviewer. In the current study, years were not the focus of the study that´s why we didn´t analyzed them in our current results. Moreover, the study is already very extensive with a large number of results, if we are adding years as another factor then it will make the study more laborous for the readers. We hope that reviewer will agree with us at this point. 

Point 6: Rewrite conclusion in one paragraph in detail.

Response 6: Thanks, the conclusions are revised and highlighted in red color.

Reviewer 3 Report

Comments to the Author

I have reviewed with interest your manuscript entitled „Mechanical chiseling and cover crops effect on common bean yield in Cerrado”.

Authors in their study evaluated the effect of mechanical soil scarification associated with successive crops and predecessors of cover crops on agronomic, productive and grain yield characteristics of common bean in NTS, being implanted twelve years ago in a low-altitude Cerrado region.

The importance of the work is well justified and contributes with important information for the understanding of main problem used in the study. In general, the writing is adequate, with the necessary components for the reader's understanding. Some adjustments are suggested to qualify the paper.

In my opinion the current version of your manuscript needs same revisions. The quality of the presentation could be improved-e.g. In general, manuscript needs small improvement.

The article suffers from a number of small mistakes, ranging from misspellings to incorrectly phrased sentences.

Some adjustments are suggested to qualify the paper:

Issues include:

General comment to the Introduction section: the introduction is written in an appropriate manner. The content of the literature review chapter is related to the research topic. Up-to-date literature references are presented in the manuscript by the author, but there are same references before 2010 – 36%.

In the chapter "Materials and Methods", the methodology is adequate, but there is a lack of information in some aspects. Some more information should be explained in the text. Please provide references on the methods used to provide the soil properties.

In the chapter "Results", the results are displayed correctly.

The “Discussion” is informative. Moreover, the Authors attempt to discuss their important results and the rest is a quotation of literature. I suggest a small reorganization of Discussion, and include more up-to-date literature.

In my opinion the Conclusions is insufficiency. It could be contain more obtained by Authors results and same recommendations for farmers and other recipient of study results.

I hope that these comments help you to make an improved the final version of the manuscript

Author Response

Response to Reviewer 3 Comments

Dear Reviewer,

We would like to express our gratitude for the reviewer who took the time to provide such a thorough review of our manuscript. We believe that the changes suggested have made our manuscript much more direct and much easier to follow.

Thanks!

Point 1: The article suffers from a number of small mistakes, ranging from misspellings to incorrectly phrased sentences.

Response 1: The manuscript is thoroughly revised. Hope this version has met the expectations of the reviewer.

Point 2: - General comment to the Introduction section: the introduction is written in an appropriate manner. The content of the literature review chapter is related to the research topic. Up-to-date literature references are presented in the manuscript by the author, but there are same references before 2010 – 36%.

Response 2: The auhtors agree with reviewer. The introduction is revised and updated with recent references, highlighted in red in list of references. The introduction no longer has any references lower than 2012 (in red).

Point 3: In the chapter "Materials and Methods", the methodology is adequate, but there is a lack of information in some aspects. Some more information should be explained in the text. Please provide references on the methods used to provide the soil properties.

Response 3: Thanks, The information are added as suggested (in red).

Ponit 4: The “Discussion” is informative. Moreover, the Authors attempt to discuss their important results and the rest is a quotation of literature. I suggest a small reorganization of Discussion, and include more up-to-date literature.

Response 4: The auhtors thanks to reviewer. Discussion is revised and updated with more information to improve this section with recent references (in red).

Ponit 5: In my opinion the Conclusions is insufficiency. It could be contain more obtained by Authors results and same recommendations for farmers and other recipient of study results.

Response 5: The conclusions are added with information suggested by reviewer (in re

Round 2

Reviewer 2 Report

Dear authors, 
        Check the manuscript language and references.
Improve the visibility of the figures.
Add some soil data (physical, chemical and biological parameters
Regard,

Author Response

Dear Reviewer,

We would like to express our gratitude for the reviewer who took the time to provide such a thorough review of our manuscript. We believe that the changes suggested have made our manuscript much more direct and much easier to follow. 

Attached is the final version of the article.    We have converted the figures into 300 DPI resolution.   We added a Supplementary Table 1. Physical attributes of the soil at different depths of the experimental area, before the installation of the experiment.   All changes made are highlighted in yellow color. Most of the corrections were to improve the English writing. 

Thanks again!

Authors.
